# Deconvolution Analysis of G and F-Actin Unfolding: Insights into the Thermal Stability and Structural Modifications Induced by PACAP

**DOI:** 10.3390/ijms26073336

**Published:** 2025-04-03

**Authors:** Péter Bukovics, Dénes Lőrinczy

**Affiliations:** Department of Biophysics, Medical School, University of Pécs, Szigeti Str. 12, H-7624 Pécs, Hungary; denes.lorinczy@aok.pte.hu

**Keywords:** PACAP, DSC, actin, cytoskeleton, deconvolution, calorimetric enthalpy, thermal unfolding

## Abstract

Actin, a key component of the cytoskeleton, undergoes significant structural and thermal changes in response to various regulatory factors, including the neuropeptide pituitary adenylate cyclase-activating polypeptide (PACAP). In this study, we applied deconvolution analysis to previously obtained differential scanning calorimetry (DSC) data to resolve overlapping thermal transitions in G- and F-actin unfolding. Our findings reveal that PACAP38 and PACAP6-38 significantly alter actin stability, increasing structural cooperativity in G-actin while reducing monomer–monomer interactions in F-actin. These thermodynamic changes suggest a potential role for PACAP in modulating actin polymerization and depolymerization dynamics, contributing to cytoskeletal remodeling.

## 1. Introduction

Pituitary adenylate cyclase-activating polypeptide (PACAP) is a neuropeptide extensively distributed in the nervous system. Initially discovered in the hypothalamus, PACAP stimulates cyclic adenosine monophosphate (cAMP) production in anterior pituitary cells and promotes hormone release [1,2,3]. It exists in two biologically active forms: PACAP38, the predominant 38-amino acid form, and PACAP27, a cleavage-amidation product [4,5,6,7]. As part of the VIP/secretin/glucagon superfamily, PACAP shares structural similarities with vasoactive intestinal polypeptide (VIP) [4,6,8,9,10], while PACAP6-38 acts as a potent antagonist, further highlighting its regulatory versatility [11].

PACAP plays diverse physiological roles, regulating pituitary hormone release [7,12], cardiovascular function [13], and neuronal activity [14,15]. It exerts neurotrophic, neuroprotective, and cytoprotective effects through anti-inflammatory, antioxidant, and anti-apoptotic mechanisms. PACAP and its receptors are essential for neurogenesis, neuronal development, and cell differentiation [16,17,18,19,20,21,22,23]. Notably, PACAP enhances actin polymerization and mitigates cytoskeletal disruptions caused by C2-ceramide in cerebellar granule cells, promoting neurite outgrowth and motility [24,25]. These findings highlight the regulatory role of PACAP in cytoskeletal dynamics and cellular behavior.

PACAP demonstrates neuroprotective effects by shielding neurons from oxidative stress, ischemia, and injuries, with endogenous levels linked to survival outcomes in traumatic brain injury [21,22,23,26,27]. Its therapeutic potential extends to neurological and psychiatric conditions such as Parkinson’s and Alzheimer’s diseases [28,29], depression [30], and anxiety [28,31], though further research is needed to develop effective PACAP-based interventions.

Given the regulatory influence of PACAP on cytoskeletal elements, exploring its specific interactions with actin filaments provides critical insight into its impact on cellular organization and function. Filamentous actin (F-actin) is a crucial cytoskeletal component responsible for maintaining cell shape, structural integrity, and processes like migration, division, and transport [32,33,34,35,36,37,38,39]. Formed by the polymerization of globular actin (G-actin) into long helical filaments, F-actin creates a cytoplasmic network that provides mechanical support and scaffolding for cellular functions [32,37]. Its dynamic remodeling, driven by polymerization and depolymerization, enables cells to respond to environmental signals. Additionally, F-actin interacts with regulatory proteins to facilitate cell adhesion, polarity, and membrane protrusions, highlighting its importance in cellular organization [39,40,41,42].

In this context, understanding the thermal behavior of actin filaments provides vital insights into the molecular mechanisms governing their stability and dynamics. Tatunashvili and Privalov showed that G-actin undergoes thermal denaturation through a multi-domain process, with at least two domains unfolding independently rather than as a single cooperative unit [43]. Similarly, Le Bihan and Gicquaud used DSC and fluorescence spectroscopy to reveal that G-actin denaturation follows a two-step, irreversible process governed by kinetic control rather than thermodynamic equilibrium [44]. These findings challenge earlier models, offering deeper insight into the complex denaturation behavior of G-actin [43,44].

Bertazzon and coworkers investigated the structural dynamics of G- and F-actin, highlighting how nucleotide binding and polymerization influence their stability. They reported a two-step denaturation process for G-actin, whereas F-actin displayed more cooperative unfolding [45]. Similarly, Lőrinczy and Belágyi demonstrated that F-actin exhibits distinct endothermic components, reflecting its greater thermal complexity and stability compared to G-actin [46]. Follow-up studies confirmed significant differences in their unfolding behaviors and myosin interactions, with F-actin showing enhanced stability and flexibility [46,47,48].

The same research group also studied actin–myosin interactions under ATP hydrolysis cycle states, uncovering structural rearrangements during myosin binding [49]. Further investigations using chemical cross-linking revealed that such modifications in F-actin altered its thermal properties, reducing cooperativity and stability, particularly in the presence of heavy meromyosin [50]. These findings underscore the intricate structural dynamics of actin and its responsiveness to binding interactions and chemical modifications across diverse conditions [45,46,47,48,49,50].

Recent studies examined the effects of cyclophosphamide (CP) on the thermal stability of G- and F-actin [51,52,53]. CP rigidified the nucleotide-binding cleft, particularly in Ca^2+^-bound F-actin, contributing to chemotherapy-induced muscle dysfunction and polyneuropathy [52]. While CP destabilized G-actin, especially at higher concentrations, F-actin was more resistant in the presence of calcium. Notably, Mg^2+^-F-actin remained unaffected, whereas inter-domain linkers of G-actin showed reduced adaptability, emphasizing its sensitivity to CP-induced structural changes [51,52,53].

While CP studies highlight the importance of external factors in modifying actin behavior, the influence of PACAP represents a distinct regulatory mechanism worth deeper investigation. Takács-Kollár and coworkers studied the effects of toxofilin and twinfilin-1 on G- and F-actin stability using DSC, fluorescence spectroscopy, and microscopy [54,55,56]. Toxofilin stabilized G-actin by increasing its melting temperature, while twinfilin-1 primarily severed F-actin filaments at low pH, facilitating depolymerization. Both proteins enhanced actin stability, but toxofilin significantly raised activation energy, indicating a stronger role in monomer sequestration, whereas twinfilin predominantly influenced filament severing under acidic conditions [54,55,56].

Our recent studies [57,58] investigated the effects of various PACAP forms and fragments on the thermal stability of Ca^2+^-G-actin and Ca^2+^-F-actin using DSC. PACAP38 reduced the denaturation temperature of both actin forms, while PACAP27 slightly increased the G-actin denaturation temperature but had no effect on F-actin. Similarly, PACAP6-38 destabilized F-actin, whereas PACAP6-27 had no impact. These patterns in calorimetric enthalpy (ΔH_cal_) suggest significant influence of PACAP on actin dynamics and its role in cytoskeletal organization and neuroregeneration [57,58].

Our group also studied the role of PACAP in actin dynamics using spectroscopic methods [59]. Vékony and coworkers showed that PACAP38 and PACAP6-38 strongly bind to G-actin, increasing anisotropy and accelerating polymerization, while PACAP27 exhibited weak binding with no effect on polymerization. Fluorescence quenching confirmed structural changes induced by PACAP binding, highlighting the significant roles of PACAP38 and PACAP6-38 in actin assembly, compared to the minimal impact of PACAP27 [59].

The interaction of PACAP with G- and F-actin offers valuable insights into cellular dynamics. Building on prior research and our investigations, PACAP has been shown to influence the cytoskeletal framework, particularly actin filaments [24,25,60,61,62,63,64,65]. By applying deconvolution analysis to our previous data, we aim to elucidate the effects of PACAP on the thermal stability and unfolding of actin, further advancing our understanding of cytoskeletal regulation and its broader implications for cellular organization and nervous system behavior.

## 2. Results

The thermal denaturation of G-actin and its complexes with PACAP derivatives is shown in Figure 1. At first glance, PACAP derivatives appear to stabilize the globular G-actin structure, as the temperature interval of the half-width of the heat flux at the maximum denaturation (unfolding) temperature is significantly reduced (from T1/2 = 14.3 °C to 5.71 °C). Interestingly, residues 6–38 and 6–27 further decreased this interval compared to their original values (6.43–5.71 °C and 8.21–6.43 °C, respectively). This suggests that the interaction (cooperativity) between the structural units of the globular protein has increased, resulting in a “more tightly packed” G-actin structure. Based on the literature data, the thermal transitions of G-actin and its complexes with PACAP can be decomposed into three thermal domains, except for PACAP38 and PACAP6-38 in this study, where additional refinement was necessary for optimal fitting.

The melting temperatures and enthalpy contributions of these domains are summarized in Table 1.

In Figure 2, the deconvolution of the average F-actin denaturation curve is presented, further detailing its thermal behavior.

The main melting temperature shifted to ~65 °C as a result of polymerization, reflecting a more stable structure; however, the effects of PACAP38 and PACAP6-38 were particularly pronounced. For F-actin and its complex with PACAP6-27, three decomposed thermal domains were observed, whereas the best fit for PACAP38 and PACAP6-38 complexes required five domains, some within the enthalpy resolution range of the equipment (~4–5%). The T1/2 half-width range was smaller for all samples compared to G-actin, confirming the structure-stabilizing effect of polymerization. Interestingly, the addition of PACAP38 and PACAP6-38 to F-actin increased T1/2 (from ~4.3 °C to 5.14 °C), indicating loosening of monomer–monomer interactions within filamentous actin. The most representative thermal data are summarized in Table 2.

For F-PACAP38, F-PACAP6-38, and F-PACAP27, the enthalpy contributions of the first thermal domain fall within the resolution power range of the equipment, requiring cautious interpretation; however, this provided the best fit for the data. F-PACAP6-27 exhibited the same denaturation behavior as F-actin and is therefore not included in Figure 2 or Table 2.

## 3. Discussion

The deconvolution analysis reveals intricate thermal dynamics of PACAP-actin interactions, resolving overlapping structural transitions and providing a more detailed thermodynamic profile compared to DSC scans. PACAP38 and PACAP6-38 consistently emerged as dominant modulators, significantly influencing the thermal stability, structural cooperativity, and flexibility of both G- and F-actin, as described in our previous PACAP-actin DSC studies [57,58]. For instance, G-actin typically exhibits three thermal domains, but PACAP38 and PACAP6-38 complexes introduced a fourth, indicating altered subdomain cooperativity. Similarly, F-actin in these complexes displayed broader thermal transitions, reflecting enhanced flexibility. Conversely, PACAP27 and PACAP6-27 exhibited minimal effects, underscoring the selective influence of PACAP derivatives. These results highlight the distinct ability of PACAP38 and PACAP6-38 to modulate actin dynamics by loosening inter-domain interactions, particularly in F-actin.

This modulation can be explained by the ability of PACAP to bind within the nucleotide cleft of G-actin, altering its internal structure and modifying the cooperativity among different subdomains. The increased total calorimetric enthalpy (Table 1) suggests that the PACAP-G-actin complex adopts a more rigid conformation compared to native G-actin. Given that F-actin undergoes continuous renewal in the cytoskeleton, our results indicate that PACAP27 and PACAP38 do not significantly affect the global stability of polymerized actin (as evidenced by unchanged total calorimetric enthalpy in Table 2). In contrast, PACAP6-38 exerts a notable influence. Additionally, both PACAP38 and PACAP6-38 alter the cooperativity of subdomains within F-actin, as reflected in the shifts in denaturation temperatures and enthalpy contributions of the thermal domains. This distinction highlights the agonist–antagonist differences between PACAP38 and PACAP6-38. Mechanistically, PACAP38 and PACAP6-38 induced notable shifts in enthalpy distributions, suggesting structural remodeling within actin subdomains. These shifts align with the role of PACAP in binding the nucleotide cleft, altering domain cooperativity, and enhancing flexibility [58,59]. These findings extend our previous DSC results [57,58] by providing deeper mechanistic insights into the role of PACAP in cytoskeletal organization.

The deconvolution analysis highlights its ability to resolve subdomain-specific responses, identifying additional thermal domains and demonstrating the roles of PACAP38 and PACAP6-38 in promoting cytoskeletal flexibility and reorganization. Hence, biologically, the pronounced effects of PACAP on actin stability and dynamics underscore its involvement in cytoskeletal flexibility, a key feature in neuroregenerative and neuroprotective mechanisms [14,15,26,27,28]. The ability of PACAP38 and PACAP6-38 to modulate actin highlights their potential roles in critical processes such as cell motility, structural remodeling, and stress response [24,25]. This reinforces the hypothesis that PACAP modulates actin dynamics through targeted structural interactions, underscoring its therapeutic potential in both physiological and pathological contexts.

Notably, PACAP has been shown to play a crucial role in neural development and cytoskeletal organization, as evidenced by its expression in the developing neural tube, where it influences neurogenesis and patterning [66]. This aligns with our findings that PACAP38 and PACAP6-38 affect actin structural cooperativity, suggesting a broader role for PACAP in cytoskeletal remodeling beyond direct actin interactions. Additionally, PACAP’s activation of the PAC1HOP1 receptor has been demonstrated to regulate multiple neurotrophic signaling pathways, including Akt-mediated cytoskeletal stabilization [67]. Given that the Akt pathway is a key regulator of actin remodeling, this suggests that the effects of PACAP on actin may not only result from direct binding but could also involve receptor-mediated signaling mechanisms. Together, these findings provide further support for the role of PACAP in cytoskeletal regulation and strengthen the link between its thermodynamic effects on actin and its broader neuroprotective functions. The deconvolution method proves to be a powerful tool for resolving overlapping thermal transitions [68,69,70,71,72,73,74,75], offering a granular view of PACAP-induced structural changes in actin. By dissecting enthalpy contributions and uncovering additional thermal domains, it complements traditional DSC analysis. However, its reliance on mathematical fitting and sensitivity to experimental conditions present limitations. Future studies incorporating complementary approaches, such as a DSC comparison with the results of our group’s spectroscopic and structural analyses published in 2024 [59], will strengthen these findings and provide a more comprehensive understanding of the regulatory effects of PACAP on cytoskeletal dynamics.

## 4. Materials and Methods

### 4.1. Sample Preparation

G-actin was purified from rabbit striated muscle following standard protocols [76,77,78] and polymerized into F-actin using a polymerization buffer containing 2 mM MgCl_2_ and 50 mM KCl. PACAP derivatives (PACAP38, PACAP6-38, PACAP27, and PACAP6-27) were prepared at a final concentration of 21 μM. For further methodological details, refer to our previous publication on PACAP-actin DSC experiments [57,58].

### 4.2. Differential Scanning Calorimetry (DSC)

Thermal denaturation of G-actin, F-actin, and their PACAP complexes was measured using a high-sensitivity DSC instrument (SETARAM, Caluire-et-Cuire, France). Samples were scanned in MOPS (3-(N-morpholino)propanesulfonic acid) buffer (pH 7.8) at a rate of 0.3 K/min. Baseline subtraction and normalization were applied to the raw thermograms. The DSC data for actin and PACAP complexes were derived from two previous publications [57,58], which served as the foundation for the current deconvolution analysis.

### 4.3. Deconvolution Analysis

Deconvolution of the DSC thermograms was performed using Microcal Origin 6 software to resolve overlapping thermal transitions. The fitting procedure was refined iteratively to minimize residual error, allowing for the identification of thermal domains corresponding to structural subregions within G-actin and F-actin. Data fitting for PACAP38 and PACAP6-38 required the inclusion of additional thermal domains, highlighting their structural impact on actin subdomains. Characteristic thermal parameters, including melting temperature (T_m_), enthalpy contributions (ΔH), and the half-width at half-maximum (T_1/2_), were extracted and compared across samples.

### 4.4. Integration of Previous Results

The thermal parameters from PACAP-DSC experiments published in our earlier studies [57,58] served as a reference framework for the current analysis. Differences in thermal stability, enthalpy contributions, and cooperativity between G-actin and F-actin complexes were reanalyzed in the context of the additional thermal domains identified via deconvolution. This approach allowed us to reconcile and extend prior findings, providing deeper mechanistic insights into the role of PACAP in stabilizing or destabilizing actin structures.

### 4.5. Data Analysis

Thermal parameters derived from deconvolution were averaged across three independent measurements and reported with standard deviations. All analyses were conducted using OriginLab Origin 2018 and Origin 6.

## 5. Conclusions

The deconvolution analysis of the effects of PACAP on actin provided critical insights into the thermal stability and structural dynamics of both G-actin and F-actin. PACAP38 and PACAP6-38 consistently emerged as dominant modulators, altering thermal profiles and increasing flexibility, while PACAP27 and PACAP6-27 showed limited effects. By resolving overlapping thermal transitions, deconvolution complemented DSC, uncovering additional thermal domains and subdomain-specific interactions.

These findings reinforce the role of PACAP in cytoskeletal organization, with implications for its neuroprotective and cytoskeletal regulatory functions. The results highlight the potential of combining deconvolution with complementary methods for future studies on cytoskeletal protein dynamics.

## Figures and Tables

**Figure 1 ijms-26-03336-f001:**
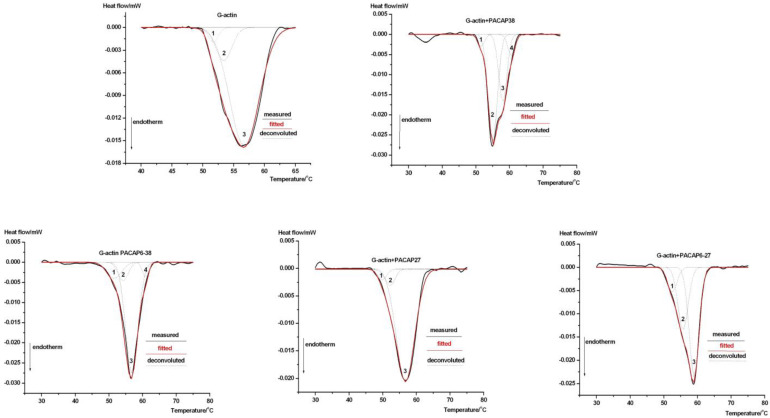
Thermal denaturation of G-actin and its complexes with PACAP derivatives. The figures represent averages of three independent measurements. The numbers indicate the possible thermal subdomains.

**Figure 2 ijms-26-03336-f002:**
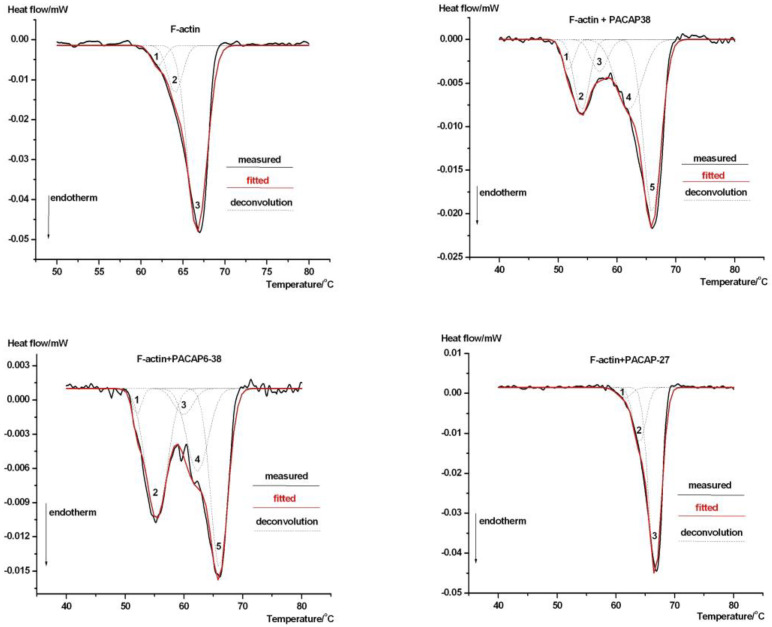
Thermal denaturation of F-actin and its complexes with PACAP derivatives. The symbols are consistent with those in Figure 1.

**Table 1 ijms-26-03336-t001:** Characteristic thermal parameters derived from deconvolution. For temperatures, data were rounded to the nearest integer above 0.05, while heat flux values include an error margin of mJ/g.

Decomposition	T_1_ (°C)	T_2_ (°C)	T_3_ (°C)	T_4_ (°C)	ΔH_Tcal_ (J g^−1^)
G-actin	51.7	53.4	56.7	-	0.023 ± 0.002
% Area	8.9	19.95	71.2	-	
G-PACAP38	51.5	55.0	58.0	60.55	0.037 ± 0.004
% Area	5.90	51.30	32.63	10.25	
G-PACAP6-38	51.5	54.2	56.8	60.85	0.038 ± 0.004
% Area	10.05	11.55	69.03	9.39	
G-PACAP27	49.8	52.2	56.75	-	0.035 ± 0.004
% Area	7.83	11.94	80.23	-	
G-PACAP6-27	52.5	55.7	59.0	-	0.031 ± 0.003
% Area	12.03	34.68	53.3	-	

**Table 2 ijms-26-03336-t002:** Characteristic thermal parameters derived from the deconvolution analysis for F-actin and its complexes. Data presentation follows the format used in Table 1.

Decomposition	T_1_ (°C)	T_2_ (°C)	T_3_ (°C)	T_4_ (°C)	T_5_ (°C)	ΔH_Tcal_ (J g^−1^)
F-actin	62.0	64.1	66.7	-	-	0.039 ± 0.002
% Area	7.88	18.77	77.34	-	-	
F-PACAP38	51.7	54.0	57.0	62.0	66.0	0.038 ± 0.003
% Area	8.09	18.84	8.55	18.50	46.24	
F-PACAP6-38	52.0	55.3	59.9	62.3	66.0	0.032 ± 0.002
% Area	5.89	29.17	5.89	18.68	40.38	
F-PACAP27	61.4	64.0	66.7	-	-	0.037 ± 0.003
% Area	4.20	22.10	73.20	-	-	

## Data Availability

Data is contained within the article.

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
