# Peer review of "Deconvolution Analysis of G and F-Actin Unfolding: Insights into the Thermal Stability and Structural Modifications Induced by PACAP"

_ijms, 2025, doi:10.3390/ijms26073336_

Round 1
Reviewer 1 Report
Comments and Suggestions for Authors
In their manuscript, Péter Bukovics and Dénes Lőrinczy investigated the effects of pituitary adenylate cyclase-activating polypeptide (PACAP) on the stability and thermal transitions of G- and F-actin. They used differential scanning calorimetry (DSC) data from two previous studies to obtain more detailed insights into the thermal transitions of the protein complexes through deconvolution of the melting profiles. The results confirmed previously obtained results and showed that two naturally occurring PACAP versions – PACAP38 and a C-terminally truncated version PACAP27 differently influence structure and stability of G- and F-actin. Deconvolution enabled a more detailed analysis of the experimental data. Since this approach to DSC data analysis is well-established and commercially available software was used, it is surprising that the Authors chose to apply deconvolution only after publishing the DSC data rather than concurrently. As a result, the manuscript does not contribute substantial new information to support a self-contained study.
Although the Authors concluded that “These findings reinforce the role of PACAP in cytoskeletal organization, with implications for its neuroprotective and cytoskeletal regulatory functions” (lines 237–238), the current data provide no evidence to support this, and in the opinion of this reviewer the final conclusions are overly speculative. The current data is rather a good starting point for a more extended study using structural or spectroscopic methods that would allow to assign the melting domains revealed by deconvolution to structural regions of G- and F-actin and their complexes with PACAP.
The Introduction is overly extensive and not well balanced with the content of the Results section. Description of previous data on actin melting behavior obtained with DSC, with particular emphasis on the work of Dr. Lőrinczy and his colleagues is too lenghty. For instance, effects of myosin and twinfilin on actin (lines 64–91) do not appear relevant to the current research. Additionally, when mentioning actin filament functions (lines 44–46), the Authors should refer to more recent review papers rather than the older work by Hanson and Lowy from 1963 (ref. 34). I also recommend avoiding references to PhD dissertations that are not readily accessible (ref. 33).
Furthermore, this study and the group’s previous work demonstrated that PACAP38 binds to actin in vitro, alters its structure, and affects polymerization kinetics. It is a very interesting observation, however, the Authors did not present or cite literature evidence confirming that direct actin-PACAP interactions occur in cells and are physiologically relevant. Notably, the information provided in the Introduction suggests that PACAP-dependent cytoskeletal remodeling is more likely mediated by cell signaling than by direct interactions with actin.
In conclusion, although the presented results seem not sufficient for publication, they are interesting. I encourage the Authors to further investigate PACAP-dependent structural remodeling of actin filaments and its potential role in the mechanisms underlying actin cytoskeleton rearrangements.
Comments on the Quality of English Language
The quality of English is adequate, but the authors could enhance the precision of their statements throughout the text.
Reviewer 2 Report
Comments and Suggestions for Authors
The article by Péter Bukovics and Dénes Lőrinczy titled “Deconvolution analysis of G and F-actin unfolding: insights into the thermal stability and structural modifications induced by PACAP” investigated the effects of the neuropeptide pituitary adenylate cyclase-activating polypeptide (PACAP) on the thermal stability and structural dynamics of G- and F-actin using deconvolution analysis of differential scanning calorimetry (DSC) data.
Comments are
- Please mention the in brief about the obtained results in the abstract.
- It will be easy for readers if you can co-relate the obtained thermal denaturation data with the actin polymerization and de-polymerization in the abstract.
- Since PACAP38 play as agonist and PACAP6-38 is an antagonist for describe their differences in the introduction.
- Please mention the ethical approval for maintaining the rabbit and the isolation of G-actin from the striated muscle in this study.
Please write the discussion simple with the obtained results relating to the polymerization.
Round 2
Reviewer 1 Report
Comments and Suggestions for Authors
The revised manuscript titled “Deconvolution analysis of G and F-actin unfolding: insights into the thermal stability and structural modifications induced by PACAP” by Bukovics and Lőrinczy still exhibits fundamental issues previously highlighted by this reviewer. All concerns raised in the initial review remain relevant to the revised version. While the Authors acknowledged the criticisms in their rebuttal letter, I would have expected them to address these issues by expanding their studies rather than merely rephrasing the existing text.
As previously noted, at this stage of investigating PACAP-actin interactions in vitro, it is essential to confirm the existence of such interactions in vivo. Since anti-PACAP antibodies are commercially available (e.g abbexa Catalogue No: abx128937), methods such as cell lysate pull-down assay, Proximity Ligation Assay, or co-localization of actin with PACAP in cells using immunofluorescence could effectively support the rationale for studying PACAP–actin interactions in vitro.
